# El Niño Enhances Snowline Rise and Ice Loss on the Quelccaya Ice Cap, Peru

Kara A. Lamantia[1,2], Laura J. Larocca[3], Lonnie G. Thompson[1,2], Bryan G. Mark[1,4]

[1] Byrd Polar and Climate Research Center, Ohio State University, Columbus, OH, USA
[2] School of Earth Sciences, Ohio State University, Columbus, OH, USA
[3] School of Ocean Futures, Arizona State University, Tempe, AZ, USA
[4] Department of Geography, Ohio State University, Columbus, OH, USA

*Correspondence to*: Kara A. Lamantia (lamantia.31.@osu.edu)

**Abstract.** Tropical glaciers in the central Andes are vital water resources and crucial climate indicators, currently undergoing
rapid retreat. However, understanding their vulnerability to the combined effects of persistent warming, the El Niño/La Niña climate phenomena, and interannual fluctuations remains limited. Here we automate the mapping of key mass balance parameters on the Quelccaya Ice Cap (QIC) in Peru, one of the largest tropical ice caps. Using Landsat's near-infrared (NIR) band, we analyze snow cover area (SCA) and total area (TA) and calculate the Accumulation Area Ratio (AAR) and equilibrium-line altitude (ELA) over nearly 40 years (1985-2023). Between 1985 and 2022, the QIC lost ~58% and ~37% of
its SCA and TA, respectively. We show that the QIC's reduction in SCA and rise in ELA are exacerbated by El Niño events, which are strongly correlated with the preceding wet season's Ocean Niño Index (ONI). Further, expansion in the QIC's SCA is observed during all La Niña years, except for the 2021-2022 La Niña. Although a singular event, this could suggest a weakened ability for SCA recovery and an accelerated decline in the future, driven primarily by anthropogenic warming.

## 1 Introduction

Tropical glaciers are important freshwater resources known to be especially sensitive to climate shifts (Kaser & Osmaston, 2002). The accelerated decline of these glaciers in response to recent warming has been widely documented over the past few decades (Bradley et al., 2006; Braun et al., 2019; Hanshaw & Bookhagen, 2014; Hugonnet et al., 2021; Pepin et al., 2015, 2022; Seehaus et al., 2020; Thompson et al., 2011, 2021; Vuille et al., 2015). In the low latitudes, glaciers are projected to lose ~69 to 98% of their 2015 mass by 2100, depending on the emissions scenario (i.e., RCP2.6 and RCP 8.5, respectively; Rounce
et al., 2023). The mass balance of tropical glaciers is strongly affected by the freezing level height (FLH), the lowest altitude in the atmosphere where temperatures reach 0°C (Schauwecker et al., 2017). In the tropics, the FLH is affected on an interannual basis by El Niño Southern Oscillation (ENSO) variations and follows the Multivariate ENSO Index (MEI) on a year-to-year basis (Bradley et al., 2009; Favier et al., 2004; Thompson, 2000; Vuille et al., 2000). Of particular concern are the effect of these transient climate events, in combination with ongoing warming, Ion the Quelccaya Ice Cap (QIC)──one of

the largest tropical ice caps in the Cordillera Vilcanota range in the outer tropical Andes. Worst case (RCP8.5) projections suggest that the QIC's 'point of no return' (i.e., the rise of the equilibrium-line altitude (ELA) above the summit) could occur as early as 2050 (Yarleque et al., 2018), leaving the QIC a wasting ice field similar to Kilimanjaro. Contemporary changes in the QIC's outlet glaciers have been frequently monitored (Brecher & Thompson, 1993) and contextualized within a longer, millennial-scale timeframe (e.g., Mark et al., 2002; Lamantia et al., 2023). For example, Mark et al. (2002) combined moraine

chronology with digital topography to model deglaciation rates during the Last Glaciation and Holocene and found that the QIC's most rapid retreat has occurred over recent centuries. Further, radiocarbon-dated plant remains from the QIC ice margin suggest that the ice cap's retracted present-day extent has not occurred in the last 7,000 years (Lamantia et al., 2023). In addition, the QIC's high-resolution ice-core records have proven invaluable for understanding past climatic and environmental variability in the region (Thompson, 2000; Thompson et al., 1985, 2013, 2017, 2021). Thus, the ongoing loss of tropical

glaciers will not only impact local communities that depend on glacial meltwater but also have implications for preserving long-term climate records, which are essential for assessing the rate and magnitude of current changes (Thompson et al., 2021).

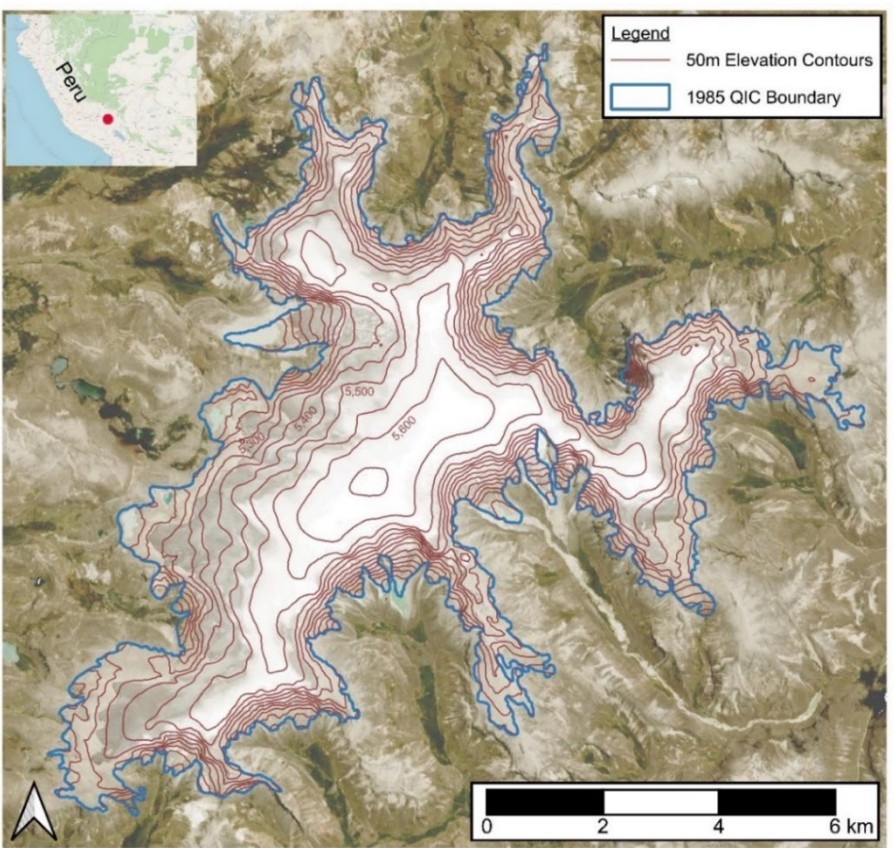

**Figure 1:** Aerial view of the Quelccaya Ice Cap (QIC; 13°56'S; 70°50'W) from October 11, 2023. The summit of the QIC reaches 5,670 m a.s.l with several outlet glaciers to the west and a steep-sided eastern portion. Base Imagery was obtained from Planet Labs Dove Satellite with 3-meter resolution and inset (top left) was obtained from the OpenStreetMap database (© OpenStreetMap contributors 2023. Distributed under the Open Data Commons Open Database License (ODbL) v1.0.)

In the tropics, there is no seasonal snow cover beyond the glacierized area that would provide an additional buffer to the ice cap's decline (Vuille et al., 2018). Quelccaya's snowfall is largely controlled by the South American Summer Monsoon (SAMS), with the snow accumulation peak in December, while moisture transport from the Amazon lowlands to the Andes is modulated by ENSO (Hurley et al., 2015). There has been no major change in hydroclimate over the last forty years in the Andes and Peruvian Amazon basin, with only 10% of stations in the Andes recording a decrease in rainfall since the 1980s

(Casimiro et al., 2013). However, ice core records from the QIC show that net accumulation in the region has been above average for the last century (Thompson, 2017). In the Andes and Peruvian Amazon basin, mean annual temperature has increased by ~0.09°C per decade over the last forty years, while maximum summer temperature records a higher increase in magnitude, ~0.15°C per decade (Casimiro et al., 2013). This warming trend is also reflected in ice core stable isotope ($\delta^{18}$O) records from multiple locations in Peru (Thompson, 2017; Thompson et al., 2013, 2017). High-resolution ice core records

indicate that the QIC is an excellent recorder of El Niño, characterized by elevated sea surface temperatures (SSTs) in the Eastern Pacific Ocean, with strong events recording isotopically enriched $\delta^{18}$O (Thompson et al., 2011, 2017). Nearby mountain ranges such as the Cordillera Blanca and Real have experienced an increase in the FLH by 160 m over the last five and a half decades (Schauwecker et al., 2017). This has implications not only for where snow can survive and accumulate (Bradley et al., 2009; Schauwecker et al., 2014; Seehaus et al., 2020) but also for the phase of precipitation and rain-snow line,

affecting surface albedo (Rabatel et al., 2013). Although the recent and past history of several tropical glaciers has been monitored and reconstructed (Brecher & Thompson, 1993; Lamantia et al., 2023; Mark et al., 2002; Vuille et al., 2018; Yarleque et al., 2018), their sensitivity and response to the combined effects of sustained warming and interannual climate variations, such as ENSO, have yet to be extensively evaluated over recent decades. The QIC is located in an ideal setting to assess the impact of these collective effects on tropical glacier vulnerability.


Since routine ground-based measurements in remote locations such as south-central Peru are difficult to maintain, using satellite imagery to estimate the ELA has become a viable option for long-term glacier monitoring. Previous studies have shown that the end of the dry season (September) location of the snowline altitude (SLA) can act as a proxy for the ELA and ultimately be used to infer the mass balance of a glacier or ice cap (Fang et al., 2011; Hu et al., 2020; Liu et al., 2021;

Racoviteanu et al., 2019). In the outer tropics, Rabatel et al. (2012) compared manual assessment of SLA on the Artesonraju and Zongo glaciers via Landsat and SPOT imagery with field-based ELA measurements and found the highest SLA during the dry season provides a good estimate of the annual ELA. More recently, Yarleque et al. (2018) used Landsat observations of the highest annual SLA to calibrate the ELA-FLH relationship to assess the future state of the QIC in response to several warming scenarios. Here, we employ cloud-based analysis of satellite imagery to assess the QIC at the end of the dry season

between 1985 and 2023. We automate not only the detection of the snow-covered area (SCA) and total area (TA), but also the calculation of the accumulation area ratio (AAR), the median elevation of the SCA, and the SLA as a proxy for the ELA. Changes in the ELA, SCA, and AAR are analyzed alongside ERA5–Land Reanalysis Climate Data from the European Centre for Medium-Range Weather Forecasts (ECMWF), including total precipitation and 550 hPa temperature, as well as multiple

ENSO Indices: the MEI, the Ocean Niño Index (ONI), and the Southern Oscillation Index (SOI). We focus our analyses on the strongest most recent El Niño events (1998, 2016, and 2023) and the QIC's response to these interannual climate anomalies.

## 2 Methods

### 2.1 Current Analysis Techniques

Manual snowline tracing is often limited to high-quality imagery to discern between snow and ice. However, recent advances in image analysis have allowed for the automation of snowline detection via satellite imagery. Typically, a suite of images, often from Landsat satellites, are paired with one or more Digital Elevation Models (DEMs) and a glacier outline within the temporal scale of interest (Li et al., 2022). From there, a variety of thresholds are evaluated and set for the area of interest to separate snow from ice, and extract the position of the transition (Racoviteanu et al., 2019). There are challenges in this process, including the adjustment of surface reflectance for varying topographies, the occurrence of patchy snow cover on the glacier surface, and highly variable atmospheric conditions that require the algorithm to be customized for the location of interest (Racoviteanu et al., 2019). Previous studies on Andean glaciers have used a handful of techniques to extract the location of the snowline and to overcome some of the aforementioned challenges, including spectral mixing analysis, simple band ratios and filtering, hillshade mask shadow removal, and manual editing (Hanshaw & Bookhagen, 2014; Klein & Isacks, 1999). Here we implement an automated approach that employs a topographic correction, followed by segmentation of the NIR band via the Otsu method (Otsu, 1975), which we describe in further detail in section 2.3.

### 2.2 Satellite Data Collection

To automate the SCA detection and ELA calculation, the following data inputs were required: an annual satellite image, a DEM, and the 1985 outline of the QIC. Using the Google Earth Engine platform we selected annual Landsat images as close as possible to September 15th with clear visibility of the QIC from 1985 to 2023 (Table S1). Mid to late September marks the end of the dry season in the Cordillera Vilcanota, which enabled analysis of the ice cap without extraneous snowfall around the perimeter. Imagery from each year was on average ±23 days within the target date and was manually inspected to ensure no recent snowfall events occurred. If September imagery was not available, October and November images were collected, and if imagery was still not available August and July images were collected with the intent to capture the closest end of dry season conditions at the QIC. No images were used if a recent snowfall event was evident. Sentinel-2 imagery was used in 2021 and 2023, due to a lack of cloudless images from Landsat 8/9. Separate scripts were adapted for each satellite (i.e., Landsat or Sentinel-2). We note that the 2023 results are not included in our initial analysis of QIC's ELA change as it is part of an incomplete El Niño event. No imagery was collected for the years 1987, 1993, 2004, 2012, and 2018 due to high cloud cover and/or visible snowfall events. We used two DEMs to account for changes in ice elevation over time and any down wasting of the QIC. The NASADEM, created from the Shuttle Radar Topography Missions, was implemented from 1985 to 2005. Post 2005, the COP30 DEM, released in 2010, was implemented. Additionally, multiple images (16 and 18, respectively)

were collected before, throughout, and after the two largest El Niño events (i.e., during the periods 1997-1999 and 2015-2017), from June of the first year to November of the last year, to assess intra- and interannual change and response of the QIC to these events.

## 2.3 Satellite Image Analysis of Snow Cover Area

To begin, the least cloudy image from the target year closest to the end of the dry season was clipped to the region of interest, the delineated QIC boundary (Step 1; Fig. S1). Pre-processing of each image included calculating the slope and aspect of the region of interest from the DEM (Step 2). We implemented the Ekstrand Correction (Ekstrand, 1996) to account for topographic effects such as shadowing due to differences in sun elevation and incidence angle (Step 3) rather than a pixel-based Minnaert Correction method (Ge et al., 2008) which resulted in the over-correction of the steeper eastern side of the QIC. To delineate the snow cover area (SCA), the NIR band was assessed with an image segmentation algorithm, the Otsu method (Gaddam et al., 2022; Otsu, 1975). This results in a bimodal frequency histogram where an automatically detected threshold separates snow from ice (Step 4; Fig. S2). Once calculated, it was applied to the NIR band to create a binary mask of snow and ice (Step 4). The annual image and DEM were then clipped to the snow mask creating the SCA, and the DEM data was extracted (Step 5). Following this, the SCA was calculated based on the number of pixels and image resolution, and the median elevation of the SCA was determined. SCAs were exported to shapefiles and the DEM data was exported as a histogram in 50-meter elevation bins (Step 6).

## 2.4 Calculation of Total Area, AAR, and ELA, and Uncertainty

As the SLA is a proxy for the ELA, we will use the term ELA from this point forward. In pursuit of the ELA, we calculated the Accumulation Area Ratio (AAR). The AAR is defined as: AAR = Ac/(Ac +Ab) where Ac is the accumulation area, Ab is the ice covered area, and Ac + Ab is the total area (TA) (Meier, 1962). In this case, Ac is the SCA and Ac+Ab is the TA (both ice and snow). To calculate the TA, we automated the calculation of the Normalized Difference Snow Index (NDSI), which leverages the reflectance of snow and ice in the green and short-wave infrared (SWIR) spectra compared to other land cover types. The NDSI is calculated as follows: NDSI $= (\rho_G - \rho_{SWIR})/(\rho_G + \rho_{SWIR})$, where $\rho_G$ and $\rho_{SWIR}$ are the reflectance of the green and short-wave infrared bands, respectively (Dozier, 1989; Hall & Riggs, 2007). We again used the Otsu thresholding method to calculate the NDSI threshold, typically set around 0.4 (Dozier, 1989; Hall & Riggs, 2007; Otsu, 1975; Sankey et al., 2015). By applying the threshold to each image, we obtained a binary image of snow and ice versus land cover and used this to calculate the TA (i.e., by multiplying the number of snow- and ice-covered pixels by the appropriate pixel resolution; Step 7). The AAR was calculated by dividing the SCA by the TA (Step 8). The ELA was calculated using the DEM and the AAR by identifying the elevation at the 1 – AAR percentile of elevations in the TA (Step 9: Fig. S3). For example, if the AAR is 0.8, we assume the ELA is located at the 20[th] percentile of elevations in the TA. In summary, for each image analyzed, we obtained the SCA, the median elevation of the SCA, the TA, the AAR, and the ELA. Calculated results for the SCA and TA via our automated methods agree with ten SCA and TA manual digitizations (±3%) performed in the following years: 1985,

1989, 1992, 1998, 2002, 2006, 2010, 2016, 2019, & 2023. Other studies have shown automated detection of snowlines produce similar results to manual digitization (Hanshaw & Bookhagen, 2014) with automated detection being preferable, as repetition is simpler and any error is likely to be more consistent (Paul et al., 2013).

## 2.5 Renanalysis Climate Data and ENSO Correlation

To compare the QIC's SCA and ELA with climate, we used daily and monthly averaged ERA5–Land Reanalysis Climate Data from the European Centre for Medium-Range Weather Forecasts , including total precipitation and 550 hPa temperature. We divided the data into wet (October to April) and dry (May to September) seasons based on precipitation records and past literature (Kaser & Osmaston, 2002; Veettil et al., 2017). To assess changes in climate at the QIC over time, we calculated average precipitation and temperature in five-year intervals, as well as the average number of days above and below freezing for each season and year from 1985 to 2023. Finally, to assess the QIC's interannual response to climatic anomalies, we paired detrended ELA, SCA, and median elevation of the SCA with the MEI, SOI, and ONI indices for correlation, obtained from the National Oceanic and Atmospheric Administration (NOAA; https://www.weather.gov/fwd/indices). We defined El Niño and La Niña periods using the ONI index. The ONI index is measured as the 3-month running mean of sea surface temperature (SST) anomalies in the Niño 3.4 region (5˚N - 5˚S and 120˚W – 170˚W). Traditionally, the anomalies must exceed the ±0.5°C threshold for at least 5 consecutive 3-month periods to classify as a "full-fledged" El Niño or La Niña. Here, we used an ONI index threshold of ±1.0°C instead of the usual ±0.5°C to isolate the strongest events across the last four decades (i.e., El Niño: ONI ≥1; La Niña: ONI ≤-1). As the target month of observation for each year was September, QIC variables for each year were correlated with ENSO indices over the preceding months (i.e., over the year before the observation date) to analyze the response of the QIC to climatic anomalies. For example, correlations were tested between QIC variables in September 1998 and ENSO Indices from September 1997 to August 1998. We evaluated all three previously mentioned ENSO indices but have chosen to focus on the ONI Index in the following sections as it presented the clearest patterns between ENSO and the assessed QIC variables.

## 3 Results

### 3.1 The Decline of the QIC and Multi-Decadal Climate Trends

Between 1985 and 2022, the QIC lost ~37% of its TA and ~58% of its SCA (1985: TA=~58.7 km$^2$, SCA=~46.4 km$^2$; 2022: TA=~36.7 km$^2$, SCA=~19.7 km$^2$; Table S2). Between the first and last five years of this observational period (i.e., 1985-89 and 2018-22), the QIC's TA and SCA declined by ~29% and ~38%, respectively. This TA loss is concurrent with a retreat of the SCA to higher elevations (Fig. 2). We observed a ~209 m and ~113 m rise of the ELA and median elevation of the SCA, respectively (1985 to 2022). In 1985, 90% of the SCA existed above 5,250 m a.s.l., and in 2021, 90% of the SCA shifted to elevations above 5,350 m a.s.l. Further, by 2022, 90% of the SCA shifted to even higher elevations above 5,400 m a.s.l. On average, the SCA and TA decreased by ~0.72 km$^2$ and ~0.59 km$^2$ per year, with an average yearly ELA rise of ~5.65 m per

year. If we consider the full observation period (1985-2023), linear regression models suggest a loss of 0.47±0.09 km$^2$ yr$^{-1}$ (R$^2$=0.44, p<0.001) in the QIC's SCA; a loss of 0.49±0.02 km$^2$ yr$^{-1}$ (R$^2$=0.93, p<0.001) in the QIC's TA; and an average rise of 3.61±0.79 m yr$^{-1}$ (R$^2$=0.40, p<0.001) in the QIC's ELA. However, the removal of the three strongest El Niño years (1998, 2016, and 2023) resulted in slower average losses in QIC's SCA, and a slower average rise in QIC's ELA: -0.42±0.07 km$^2$ yr$^{-1}$ (R$^2$=0.58, p<0.001); and +3.25±0.64 m yr$^{-1}$ (R$^2$=0.47, p<0.001), respectively (Table S3). The QIC's average AAR (not including El Niño and La Niña years) is 0.74 over the study period. For comparison, during the strongest El Niño years (1998, 2016, and 2023) the QIC's AAR was lower than average (0.32, 0.40, and 0.52, respectively) while during the strongest La Niña years (1999 and 2011) the QIC's AAR was higher than average (0.83, and 0.82, respectively).

Daily and monthly variations recorded at the QIC summit and bottom margin weather stations from Bradley et al. (2009) are well correlated with the ERA5-Land 550 hPa temperature dataset (R$^2$= 0.81), which was used to determine changes in temperature throughout the observation period. Between the first and last five years in our observational period, the reanalysis climate data show a ~0.60°C increase in wet season (October–April) temperature and a ~1.14°C increase in dry season (May–September) temperature. Correspondingly, the number of days above 0°C rose from 1% to 6% in the wet season and from 0.5% to 8% in the dry season between the first five and last five years. These results are consistent with previous studies that suggest a ~0.1°C per decade rise in upper air temperature, and a rise in the height of the freezing level (~45 m between 1977-2007) in the tropics near the QIC (Bradley et al., 2009; Vuille et al., 2008). We observe no significant change in precipitation in either the wet or dry seasons (wet: R$^2$=0.02, p=0.11; dry: R$^2$=0.03, p=0.09), and on average, 73% of the precipitation occurs during the wet season.

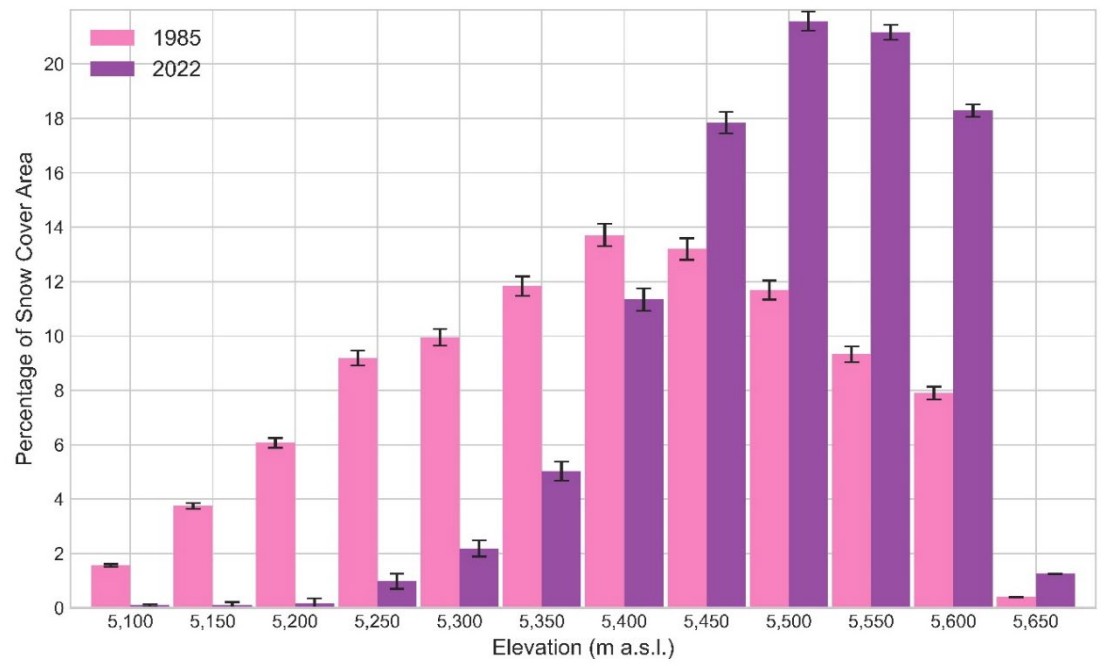

**Figure 2:** Percentage of snow cover area (SCA) in 50-meter elevation bins, demonstrating the shift of the SCA to higher elevations. Error bars represent ±3% uncertainty calculated from comparisons to manual digitization.

### 3.2 Response of the QIC to Interannual Climate Phenomena

The strongest El Niño events (1998, 2016, and 2023) coincide with a large decrease in the QIC's SCA. We observed a ~59% reduction in SCA from 1997 to 1998 and a 49% reduction from 2015 to 2016. Likewise, the QIC's AAR decreased from 0.71 to 0.31 from 1997 to 1998, and from 0.76 to 0.41 from 2015 to 2016. In 1999 (the year following the 1998 El Niño), the QIC's SCA fully rebounded back to 1997 conditions. However, in 2017 (the year following the 2016 El Niño) the SCA only reached about 77% of its 2015 value (2015 SCA=~32.4 km$^2$; 2017 SCA: ~24.9 km$^2$; Fig. 3). A modest increase in the SCA is observed

in 2019 (no imagery was available for 2018), however, to date the SCA has not returned to its pre-El Niño 2015 extent and has continued to decline through 2022. The 2023 measurements occur during an ongoing El Niño event. However, if we consider the additional 2023 El Niño year, between 1985 and 2023, we observe a ~61% decline in the QIC's SCA in just under 40 years (Fig. 4). While the height of the 2023 El Niño did not occur until December of 2023 (ONI = 2.0), by our definition an El Niño was in effect as of July 2023 (ONI = 1.1), and measurements were collected in October 2023 (ONI = 1.8). The

QIC's SCA in 2023 was ~17.97 km$^2$—a 9% loss compared to that of 2022. The 2023 AAR was 0.53, well below the average (i.e. 0.74), and from 2022 to 2023, we observed a 16 m and 8 m rise of the ELA and median elevation of the SCA, respectively.

Overall, the ONI Index is most strongly correlated with the median elevation of the SCA (Fig. 5), its Pearson coefficient from the preceding April back through the previous September ranging from 0.46 to 0.61 (p<0.05; Table S5). The ONI Index and ELA are similarly positively correlated (0.41 to 0.58 April-September), while the ONI Index and SCA exhibit a negative

correlation of similar strength from April through September (-0.44 to -0.60). To better assess the QIC's response to El Niño events, we utilize our high-frequency (monthly) observations collected around the 1998 and 2016 El Niño events (i.e., between 1997–1999 and 2015–2017). We found that during both the 1997–1999 and 2015–2017 periods, the lowest SCA occurred during the mid-September observations of El Niño year (1998 and 2016), after a decline in SCA that began from the previous year's September measurement (1997 and 2015). The ENSO indices are most strongly correlated with the QIC's ELA, SCA,

and median elevation as they best represent the changing ice distribution and mass.

In addition, we found that linear regression models for ELA and SCA that include El Niño as a binary predictor (i.e., yes or no) improve the R$^2$ values from 0.40 to 0.67 (p<0.001) and 0.44 to 0.72 (p<0.001), respectively, while the R$^2$ value for the model predicting TA does not improve with the inclusion binary predictor. Finally, to compare the yearly means of the QIC

variables over the full observational period, we characterized each year as El Niño, La Niña, or neutral using the ONI Index (El Niño $\geq$ 1; La Niña $\leq$ -1; neutral= -1 < ONI < 1) and conducted an ANOVA with a post hoc test. We found a significant difference in the mean SCA and ELA between El Niño and neutral years, in the mean AAR between El Niño and neutral years, and El Niño and La Niña years (Fig S4). There is no significant difference in the mean TA between El Niño, La Niña, and neutral years (Table S4). Together the improved R$^2$ values and ANOVA results further indicate that El Niño events have a

substantial effect on QIC's yearly SCA and related variables, but not the TA. During El Niño events, ERA5-Land climate data show a marked increase in air temperatures while precipitation patterns and magnitude remain largely unchanged. For instance, during the dry season from 1997 to 1998, there is a ~1.35°C increase in temperature and a change in total precipitation of less than ~0.02 meters. This suggests that the reduction in QIC's SCA is likely primarily driven by increased temperatures during the events, rather than by reduced precipitation. However, changes in the dominant phase of precipitation (e.g., more rain

versus snow) could also be a contributing factor, but this is beyond the scope of this paper.

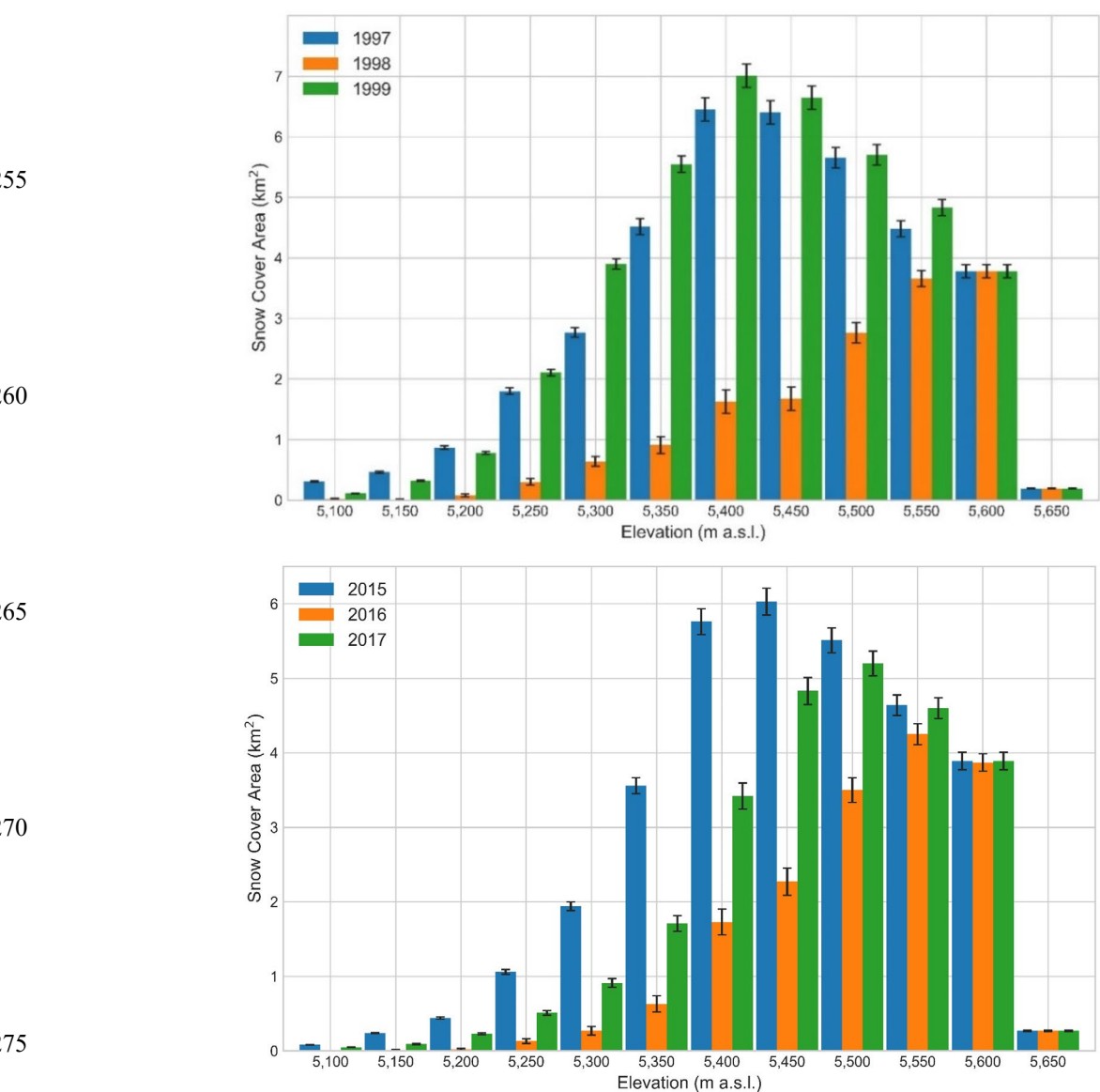

**Figure 3:** Distribution of snow cover (km²) showing the reduction and rebound of the SCA during and following the 1998 El Niño event (top) and reduction and incomplete recovery of the SCA during and following the 2016 El Niño event (bottom). Error bars represent ±3% uncertainty calculated from comparisons to manual digitization.

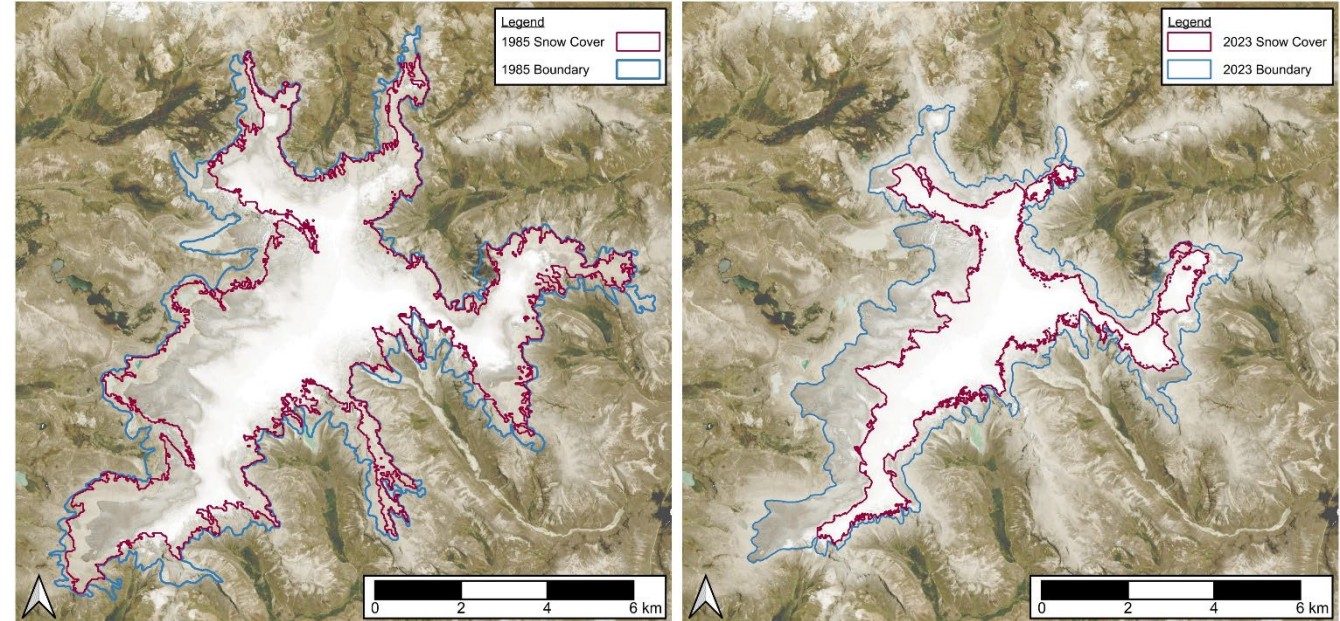

**Figure 4:** Decrease in the QIC's SCA (red) and TA (blue) at the end of the dry season between 1985 (left) and 2023 (right). Base Imagery obtained from Planet Labs Dove Satellite with 3-meter resolution, October 2023.

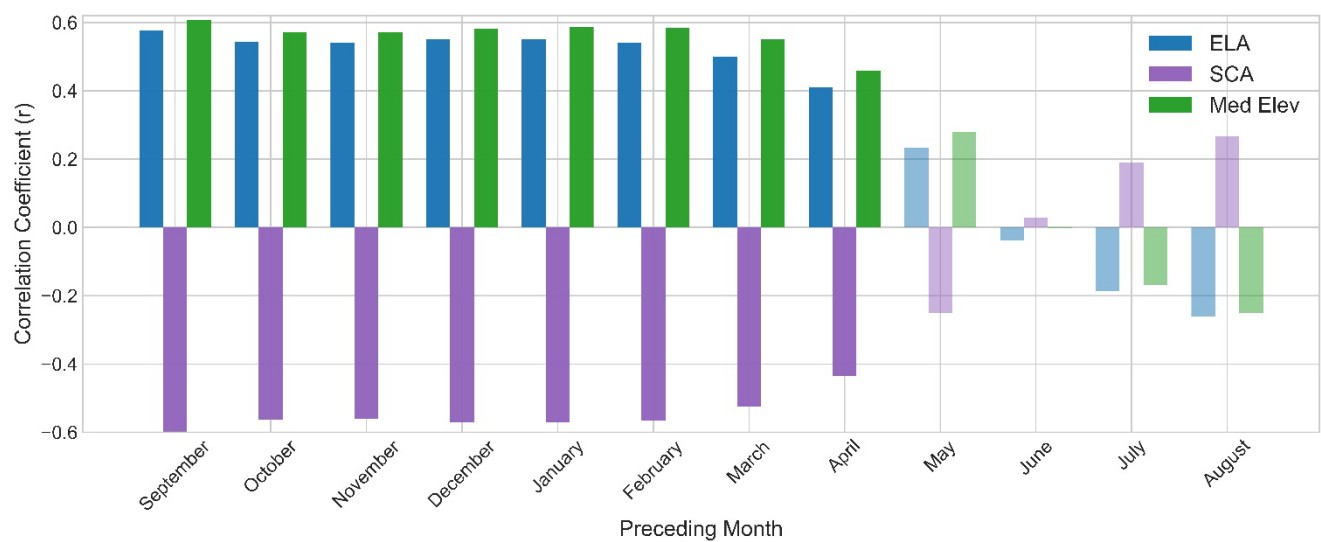

**Figure 5:** Zero-order correlations ($r$) for QIC variables (ELA, SCA, and median elevation of the SCA (Med Elev)) and the ONI Index from 1985 to 2023. Correlation coefficients with non-statistically significant p-values ($>0.05$) are denoted as semitransparent bars.

## 4 Discussion

### 4.1 QIC Response to Interannual Climate Variability

During El Niño events the Peruvian Andes are often drier than average (Sulca et al., 2018), with on-site measurements at the QIC recording warmer and drier conditions (Hurley et al., 2019). To be considered an El Niño event, the SST anomalies must be high for at least four consecutive months (Lagos et al., 2008). These events are evident in QIC shallow ice cores which display a 'smoothed' $\delta^{18}O$ signal during El Niño instead of the usual high-resolution variability, indicating dry and warm conditions leading to a lack of accumulation and increased melt (Thompson et al., 2017). Our results suggest these multi-month positive temperature anomalies have a greater influence on the QIC's SCA than precipitation (i.e., no significant correlation is observed between wet season precipitation and the SCA, ELA, or SCA median elevation). We demonstrate that the El Niño events in 1998 and 2016 correspond to large reductions in QIC's SCA (Fig. 6). These SCA perturbations are outliers from the mean SCA (z-score= -2.3 and -2.11, respectively; average z-score = 0.12). Linear regression analysis with and without El Niño event years show differing slope coefficients, indicating that these events are also associated with an enhanced reduction in QIC's SCA over the full observational period. As noted in the results, the SCA rebound in 2017 from the 2016 El Niño minimum, only reached about 77% of its 2015 value. Whereas, in 1999 (following the 1998 El Niño minimum), the SCA fully recovered, above its 1997 value. The year 1999 was one of the strongest La Niña events (ONI Index of -1.5) within the observation period (along with 1989 and 2011). This suggests that the timing and magnitude of La Niña events represent an additional important factor influencing the interannual variability of the SCA.

We observe a strong and significant decline in the QIC's TA over the observational period: $-0.49 \pm 0.02$ km$^2$ yr$^{-1}$ ($R^2$=0.93, p<0.001). However, comparisons between linear regression models for QIC variables with and without the inclusion of El Niño as a binary predictor suggest that El Niño years have a stronger effect on SCA, AAR, and ELA than on TA (Fig. S4, Table S4). This indicates that while the QIC's SCA is notably briefly reduced, and its decline exacerbated over the long term, by El Niño events, anthropogenic warming is the primary driver of the multi-decadal decline of the QIC's SCA and TA (Bradley et al., 2009; Rounce et al., 2023; Thompson et al., 2021; Vuille et al., 2018; Yarleque et al., 2018). Further, as previously noted, the QIC's SCA experienced some level of temporary expansion during most La Niña events over the last 40 years (Fig. 6). However, during the 2021-2022 La Niña, the SCA declined. While this represents only a single incident, this behavior may persist with the onset of the predicted 2024/2025 La Niña, as the QIC continues to be increasingly impacted by the combined effects of El Niño events and anthropogenic warming. Another consequence of overall warming, the reduction in the percentage of days at or below freezing during the wet season, along with a rise in the FLH, will further exacerbate the decline in QIC's SCA. In addition, a recent study has projected a faster onset and slower decline of future El Niños (Lopez et al., 2022). Together these effects will act to further reduce the QIC's SCA and to enhance mass loss.

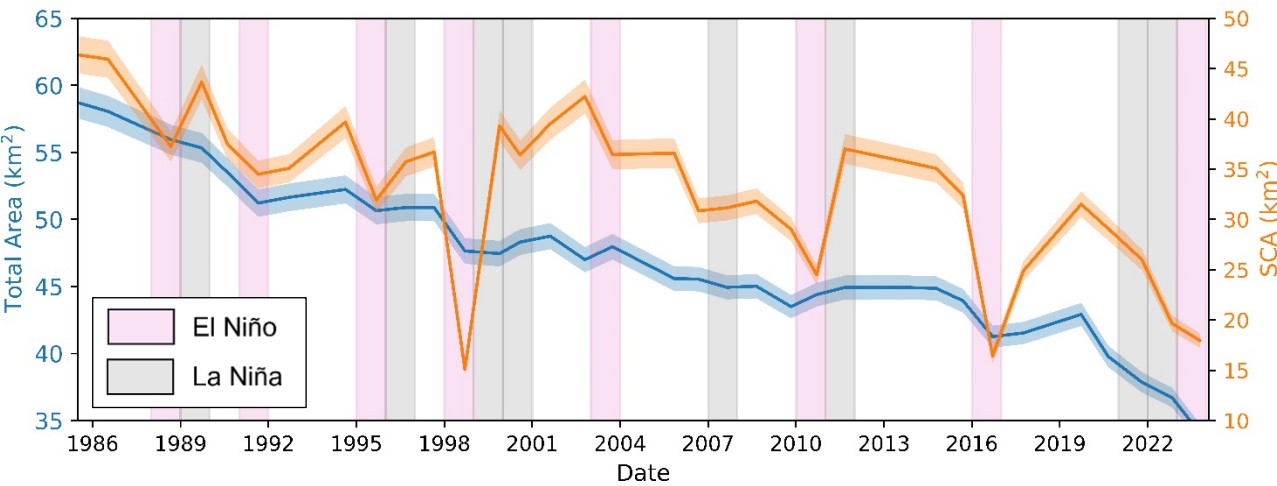

**Figure 6:** Decline of QIC's TA and SCA over the observational period (1985-2023). The timing of El Niño and La Niña events (with years containing ONI indexes that exceed the threshold of ±1.0) are noted in pink and gray colored bars, respectively. Shading around the TA and SCA lines represents ±3% uncertainty calculated from comparisons to manual digitization.

### 4.2 Steady-State of the QIC and comparison with other studies

For glaciers in other locations, such as the New Zealand Alps and the European Alps, the steady-state AAR (associated with zero annual net mass balance) is around 0.6 (Benn & Lehmkuhl, 2000), but for tropical glaciers, the steady-state AAR is higher, ~0.8 (Kaser & Osmaston, 2002). Discounting El Niño years (1998, 2016, and 2023), the QIC's average AAR is 0.74, indicating the QIC is out of equilibrium, and likely somewhat lagging in response compared with the pace of ongoing climate change. The ice cap is pushed even more noticeably out of equilibrium during the observed El Niño events with AARs of 0.31, 0.41, and 0.53. These are far lower values than required for even high latitude glaciers, and far below the average for the QIC. Our results indicate the SCA has changed more dramatically on a year-to-year basis than the TA (Fig. 6), indicating rapid response of the SCA and thus the ELA to interannual climate variability, in addition to decadal-scale changes (Zekollari et al., 2020). This is consistent with other studies that indicate the QIC is likely to respond to climate drivers within a few decades from the present, including the almost immediate response to El Niño events (Thompson, 2017; Veettil et al., 2017). Previous work on the QIC indicated the median elevation of the entire QIC rose ~1.59 m per decade from 1975 to 2010 (Taylor et al., 2022), which is slightly less than our estimate (~1.91 m per decade), although we note the differing temporal periods. Similarly, previous studies of the QIC note a mean ELA between 1992 and 2017 of ~5,436 m a.s.l. (Yarleque et al., 2018) while our automated methods suggest a lower mean ELA of ~5,351 m a.s.l. for the same temporal period. A linear projection of the ~40-year trend in SCA and TA suggests that the QIC could lose its SCA before 2080 (becoming a wasting ice field) and may completely melt away before 2100 (Fig. S5). However, we suggest that these simplistic linear projections are conservative. With the increasing loss of the SCA and further warming, there is potential for uneven surfaces with standing water, and increases in rainfall to alter surface albedo (Naegeli & Huss, 2017; Wang et al., 2015). In addition, as ice caps shrink and become thinner, elevation-dependent feedbacks and edge effects become increasingly important, resulting in accelerated

shrinking over time, especially given the large flat topography making up most of the QIC's remaining ice-covered area. Thus, these combined effects are likely to accelerate the QIC's decline.

**5 Conclusion**

Here we automate the process of satellite-based collection of yearly variables important for mass balance assessment on the QIC and evaluate the ice cap's response to interannual climate fluctuations in combination with multi-decadal climate changes.

We observe a decline in TA, SCA, and a rise in ELA over the last ~40 years, as well as high interannual variability in SCA and ELA, correlated with ENSO events. Specifically, we observe a ~42% loss in TA, a ~61% loss in SCA, and a ~225 m rise of the ELA from 1985 to 2023. The QIC's SCA at the end of dry season, is significantly correlated with the ONI at the height of the previous wet season, with marked decreases in the SCA and AAR during El Niño events. While the SCA has rebounded in response to La Niña events in the past, the SCA has declined through the most recent La Niña. Continued monitoring of the

QIC will be vital, as the potential for various surface processes and future El Niño events to accelerate QIC's ice loss will rise with continued warming. Further, the QIC's future demise points towards water scarcity for the local population, creating uncharted difficulties, especially seasonally (Veettil et al., 2017; Vuille et al., 2018).

**Code and Data Availability**

All calculated QIC variables from this study are provided within the supplementary information, detailed in Table S2. Annual SCA shapefile data and DEM bin distribution initially calculated within Google Earth Engine are available at the following repository at https://doi.org/10.5281/zenodo.11265568. A sample code for preprocessing and processing Landsat 8 images is available at the following url: https://code.earthengine.google.com/cfcbd0780ff3f09b0698035cd6dd678a.

**Author Contribution**

K.A.L. designed the study, developed the code, collected the snow cover data, and completed the analysis of the data and accompanying climate variables. K.A.L. wrote the manuscript. K.A.L., L.J.L், L.G.T., and B.G.M. contributed to the discussion of the results, editing, and revision of the manuscript.

**Competing Interests**

The authors declare that they have no conflict of interest.

**Acknowledgments**

This research was supported by the Heising-Simons Foundation and Volo Foundation for both past field data used as reference

and in support of the current project. We would like to thank the National Science Foundation (NSF) for graduate student

support under Award #1805819. Additionally, we thank James Lea for providing Google Earth Engineassistance with topographic corrections, Rainey Aberle for providing guidance regarding ELA calculations, and Shelby Turner for insights into their climate projections in the Peruvian Andes. This is Byrd Polar and Climate Research Center contribution No. 1630.

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
