# Peer review of "El Niño Enhances Snowline Rise and Ice Loss on the Quelccaya Ice Cap, Peru"

_EGUsphere, 2024_

## Referee Comment (RC2)

This manuscript evaluates the change in the Quelccaya Ice Cap from 1985 to 2023, using satellite imagery. The year-to-year changes are described, and an evaluation is conducted as to the change in strong El Niño years. The analysis is interesting, particularly the discussion of the snow covered area rebound time after El Niño events.

Some clarification is needed before publication, particularly on the definition of El Niño years and some of the statistical analysis. Detailed comments are provided below.

**Comments**

Full manuscript: Could the authors clarify which El Niño years are chosen for which parts of the analysis in this manuscript, and why? The authors focus on the 1997-1998, 2015-2016 and 2023 events for much of the manuscript, but use more years for figure 5, for example. Some graphs (e.g. S4) are unclear in which ENSO years are used.

Line 86 and Table S1: The authors have picked the closest cloud-free images to September 1$^{st}$ in order to calculate the area of the ice cap accurately, however these images vary from June to October. Can the authors comment on the effect of this difference on their results?

Line 96: 1999 was a strong La Niña year, not an El Niño year (as the authors note in line 261). 2017 was a neutral year. Can the authors clarify the dates of the 16 and 18 images collected, and what the ONI index was for the rolling three month period covering those dates?

Line 126: Could the authors expand on the difference between the manual calculation of ELA and the automated calculation? Is the 3 % in this instance 3 % of the elevation above sea level? If so, this would be greater than the change in ELA between 1985 and 2022 that the authors calculate on line 143, and may require further consideration.

Line 142-146: The authors have calculated the loss of TA and SCA based on the first and last years of data only. Especially for SCA, the year-to-year variation is relatively high. I would suggest the authors calculate the loss through the average loss (which would then match with the loss calculated through linear regression), or take the average of the first few and last few years. This comment also holds for figure 2.

Line 154: Could the authors clarify why they have removed El Niño years to calculate the trend in AAR, but not comparative La Niña years? The analysis shown in figure S4, which clearly shows the difference between El Niño, La Niña and neutral years, may be more appropriate to determining the effects of El Niño events on the AAR.

Figure 5: The x-axis seems to be misaligned in this plot. For example, the lowest SCA occurred in 1998, along with the El Niño year. Both of these are plotted covering the tick mark for 1999.

Line 163: I agree with the other reviewer comment that it is necessary to apply a simple bias-correction to the ERA5 temperature, to account for the difference in elevation between the QIC and the ERA5 elevation, in order to determine the percentage of days above the freezing level height. I would also suggest that the authors' findings of an increase of 1.24 degrees in the dry season over 37 years is somewhat greater than the 0.1 degree/decade in the quoted references.

Line 187 and figures 2 and 3: These figures might be easier to interpret if they were plotted the total SCA in each elevation bin, rather than the percentage. For example, in figure 3 (lower), the

percentage of SCA in 2015 and 2017 is similar at low elevations, suggesting the SCA may be similar in both years, but they vary considerably at higher elevations. Total SCA would then match better with the text stating the change in SCA is 70% of its 2015 value.

Line 253: Lagos et al., 2008 record higher than average precipitation during El Niño events along the northern coast, but a mixed response in the Amazon and the Andes (including around the QIC).

Line 263: Could the authors explain how they have calculated the drop in correlation from 1998 to 2016? Are these correlations based on the yearly measurements and yearly ONI? If so, how is the correlation calculated for a single year? Similarly, how are the regressions calculated in line 266 (and what are the multiple variables used?).

Line 265 and 297: the authors state that anthropogenic warming is overwhelming natural climatic signals, but in this manuscript, the most severe El Niño effects seem larger than the overall reduction from anthropogenic warming (e.g. figure 5, 1998/1999)

**Minor suggestions**

Line 9: I would suggest rephrasing the first sentence to "Tropical glaciers in the central Andes are vital water resources….". As it stands, water resources are mentioned twice.

Line 30: Please defined what the modern rate is of (retreat, volume  loss etc).

Line 30: 'further evidence of the QIC …'  what's this further evidence of, is it retreat?

Line 32: 'since 7000 years ago' -> 'in the last 7000 years'

Line 42: 'southern wet outer tropics' 'wet' is perhaps a typo here?

Line 48: typo 'recording documenting'.

Line 114: Please define 'Ab'

Line 152: as the AAR is defined based on the SCA and TA, I think the correlation between these variables seems inevitable.

---

## Author Comment (AC1)

**Response to Reviewers**

*We thank both reviewers for their constructive feedback. We have addressed all major and minor comments below, our responses are in blue text. Specifically, we adjusted the topographic correction within the analysis code and adjusted our target date for imagery based on comments from the reviewers. All results have been updated as well as the corresponding figures both in text and in the supplement. A note that Figure 5 is now Figure 6 in our revisions. We were informed that the colormap on Table 1 would not be possible to do with color shading so we have created a bar chart figure instead, which is now Figure 5.*

*Reviewer #1*

This is an interesting paper analyzing the relationship between ENSO and snow-covered area and ELA on the Quelccaya ice cap in Peru. The paper, however, has a number of weaknesses that need to be improved before it is ready for publication. I have tried to outline some of the main aspects that could benefit from more attention below.

**Main comments:**

Title: I think the title may have to be changed – as far as I know Quelccaya is no longer the world's largest tropical ice cap (see Kochtitzky et al., 2018). This statement is also repeated several times in the text.

*We have eliminated or adjusted mentions to resolve this comment. Our proposed new title – 'El Niño Enhances Snowline Rise and Ice Loss on the Quelccaya Ice Cap, Peru.' Line 12 – changed mention to one of the largest. Line 26 – deleted mention*

Line 22-24: Statements such as this one regarding projected future loss of a glacier surface area need to include a date, as the percentage loss quoted is time-dependent. Are you referring to the year 2100?

*Line 22-23 – Yes, the reference does refer to the year 2100. We have adjusted the mentioned sentence accordingly to read 'In the low latitudes, glaciers are projected to lose ~69-98% of their 2015 mass by 2100, depending on the emissions scenarios RCP2.6 and RCP 8.5 (Rounce et al., 2023).'*

Line 28: Yarleque et al. (2018) did not state that the QIC would disappear by 2050 under a high emission scenario. They only determined that the ELA would move above the summit by that point. Given the ice thickness of the QIC, it would likely still take several decades to fully melt all the ice in the ablation zone. Hence 2050 is a date for a 'point of no return' with the accumulation zone gone, but it is not a date for the complete disappearance of the QIC.

*Line 27-30 – Adjusted to reflect that Yarleque et al., (2018) projects the QIC's point of no return and not total disappearance. We added a mention that once the QIC is not receiving accumulation it will cease to be an ice cap with outlet glaciers and simply a wasting ice field like Kilimanjaro. Our new text reads 'The decline of the Quelccaya Ice Cap (QIC; Fig. 1), located in the Cordillera Vilcanota (CV) range in the outer tropical region of the Andes, is one such concern with worst case (RCP8.5) projections suggesting the 'point of no return' (i.e., the rise of the ELA above the summit) as early as 2050 (Yarleque et al., 2018), leading to the QIC's classification as a wasting ice field instead of an ice cap, similar to Kilimanjaro.'*

Line 32: I think what you refer to here is the actual size of the ice sheet, not the 'magnitude of retreat', which suggests something different (the loss of ice per unit time). Also the citation for this statement (Lamantia et al., 2023) is not included in the reference list

*Line 32 – Adjusted the wording 'magnitude of retreat' to 'margin-extent' to better reflect the conclusions of Lamantia et al., (2023).*

*Line 426- Lamantia et al., (2023) has been added to the reference list.*

Line 42-44. The snowfall on Quelccaya has no dynamical connection to the ITCZ. The ITCZ is a maritime feature located north of the equator. Snowfall on Quelccaya is fueled by the South American summer monsoon, with much of the heavy snowfall associated with convective activity over the western Amazon basin, triggered via cold air incursions (see Hurley et al., 2015).

*Line 66-68 – Sentence rewritten to include the Hurley reference and discussion of SAMS and ENSO as dominant factors affecting QIC snowfall and moisture transport. Now reads ' Quelccaya's snowfall is largely controlled by the South American Summer Monsoon (SAMS) with the snowfall peak in December and moisture transport from the Amazon is influenced by ENSO variations (J. V. Hurley et al., 2015).'*

Lines 50-51: The change in the FLH is not just worrisome because of direct melt, but also because it leads to a rise of the rain-snow line, thus affecting the albedo in the ablation zone. This likely has a larger impact on the total glacier energy and mass balance than the change in the sensible heat flux alone (e.g. see discussion in Rabatel et al. (2013)).

*Line 72-75 –Added the reference to Rabatel et al., (2013) and expanded sentence to read 'Nearby mountain ranges such as the Cordillera Blanca and Real have experienced an increase in the freezing level height (FLH) by 160 m over the last five and a half decades with implications for not only where snow can survive and accumulate (Bradley et al., 2009; Schauwecker et al., 2014; Seehaus et al., 2020) but also increased albedo in the ablation zone influenced by a rise of the rain/snow line (Rabatel et al., 2013).'*

Lines 52-53: Your statement here would imply that both La Nina and El Nino events lead to warmer SST in the tropical Pacific. Of course this is the case only for El Nino, while La Nina events are associated with colder SST.

*Line 76-77 – Removed mention of La Nina in this sentence to refer to only El Nino, as the ice core signature is smoothed from the warmer SSTs (El Ninos).*

Line 60-64: The first study to test this idea regarding end-of-the-dry-season snowline serving as an estimate of the ELA in the Andes was by Rabatel et al. (2012). This should be acknowledged. Their methodology was then applied by Yarleque et al. (2018) to estimate the interannual ELA variability on Quelccaya, equally relying on Landsat data to estimate the ELA via maximum elevation of the dry-season snowline. Hence the ELA approach used here has been applied on QIC before and the results obtained here should thus be compared to those previously published in Yarleque et al. (2018) to the extent that this is possible, especially when discussing the ELA results from your study in section 3.1 or during the discussion of the results in section 4.2.

*Line 88-91 – We have added mentions of both the Rabatel and Yarleque studies to read 'Initial studies in the Andes involved a manual assessment of the Artesonraju and Zongo glaciers via Landsat and SPOT imagery compared against field measurements (Rabatel et al., 2012). Yarleque et al., (2018) most*

*recently analyzed the QIC's response to warming scenarios based on the FLH/ELA relationship and future ELA projections.' Will further discuss Yarleque's results on the QIC in the discussion section 4.2. See below.*

*Line 340-344 – We have added a reference to Yarleque's findings of the QIC's average ELA compared to our calculation (within 5m) to read 'Similarly, previous studies of the QIC note a mean ELA between 1992 and 2017 of ~5,436 m a.s.l. (Yarleque et al., 2018) while our automated methods suggest a mean ELA of ~5,351 m a.s.l. for the same temporal scale. Considering the QIC's out of equilibrium state, as well as continued decline of the SCA and rise of the ELA due to ongoing anthropogenic climate change, we suggest the QIC may be completely melted away prior to 2100 (assuming the rate of loss is constant; Fig. S5'.*

Line 135: I am not sure you can use surface temperature from ERA5 directly to calculate days with above- or below-freezing on QIC without making some bias adjustments. What is the absolute surface elevation of Quelccaya in ERA5? I assume it is considerably lower than 5670 m, hence the ERA5 surface temperature will have a warm bias, no? Alternatively, you may want to use the free tropospheric temperature interpolated to the QIC elevation as it is less affected by a topographic bias than surface temperature.

*Line 162-164 – We have clarified the data usage, it was two-meter temperature from ERA5 which does take elevation into account but is slightly warm biased as the reviewer suggested. Considering most of the QIC is likely between 500-600mb of pressure we instead have used the 550mb temperatures from ERA5 daily temperature on pressure levels. This is comparable to temperature noted in Bradley et al., (2009) and we have added this reference as well on Line 186 to read 'Daily and monthly variations recorded by the QIC summit and bottom margin weather stations from Bradley et al., (2009) are well correlated with the ERA5 550mb temperature dataset, which was analyzed to determine changes in climatic variables through the observation period.'*

*Line 189-196 – The changes recorded by the 550mb temperature data have been adjusted accordingly in the results section.*

Line 185: Snow cover, unlike ice cover, varies interannually. Hence I would not refer to the change in snow cover from one year to the next as a 'loss', which sounds as if it were permanent. Maybe refer to snow cover 'change' or 'reduction' instead, but it is not a permanent 'loss'.

*Line 218-219 – Have changed the term for 'SCA loss' to 'SCA reduction' as suggested when discussing year to year change. We've also changed loss to reduction in the Figure 3 (Line 287) caption to reflect this comment. Other changes from "loss" to "reduction" are on Lines 15, 302, & 306. We have kept the term loss when referring to the entire temporal scale or yearly avg loss as it is a downward trend.*

Line 253-254: While the northern Peruvian coast indeed receives more precipitation during El Nino, the Peruvian Andes and the western Amazon are drier than normal. For example, see detailed analysis of precipitation effects of ENSO in Peru by Sulca et al. (2018). Also Hurley et al. (2019) have investigated the ENSO influence on snowfall amount and temperature anomalies on Quelccaya using on-site meteorological measurements, showing that conditions on the QIC are warmer and drier than normal during El Nino events.

*Line 297-298 – We have adjusted the introduction to the discussion section to include the above mentioned references and to properly reflect the changing meteorological conditions at Quelccaya during El Ninos. It now reads 'During El Niño events the Peruvian Andes are often drier than average (Sulca et*

*al., 2018), with on-site measurements at Quelccaya recording warmer and drier conditions (J. Hurley et al., 2019).'*

Figures 2-5: I think all these bar and line graphs require uncertainty estimates. While there is some discussion of error estimates in the text, the Figures do not include any such uncertainties. Figure 5 in particular is strange. While it make sense that the total snow covered area varies from year to year in response to ENSO, why would the total area (ice cover) increase by several km^2 from one year to the next? This would imply a rapid advance of the ice cap increasing the total ice covered area by ~10% over the course of a year, which is highly unlikely. To me this rather suggests that significant uncertainties exist in the yearly estimates of total ice cover. This makes including error bars all the more important, to understand whether interannual variations of the total ice cover reside within these uncertainty estimates.

*Figures 2, 3, & 6 (previously 5) have been edited to include error estimates (error bars on the bar charts and a semi-transparent fill behind the line graphs for figure 6). Figure 4 is updated with the most recent results. Figure 6 also displays Nino and Nina events with ONI over 1.0 for clarity. Sample below of updated Figure 2 and Figure 6.*

[Figure]

*Additionally, we have been working on improving the topographic correction and have since updated that portion of the code. This results in a much more consistent TA calculation and is reflected in new results*

*calculations and the corresponding figures. Since the end of the dry season is the middle of September to the end of October, we shifted our target date to September 15th instead of the 1st and re assessed which imagery would be best. This results in more imagery in October and less in July/August to enable the assessment of a proper end of dry season state. We also removed the years 1993 and 2018 as the only imagery available of the QIC was after a very obvious snowfall event. Reviewer #2 made a comment about the time of imagery selection, more comments are below and changes are reflected in the text on imagery selection on Lines 114-117. Description of the new topographic correction in the methods section is on Line 133-136.*

**Minor suggestions for change:**

Line 9: Glaciers are 'vital water resources as vital water resources'? Something went wrong here.

*Line 9 - We have adjusted the sentence to remove the double mention and accounted for reviewer 2's phrasing comment as well.*

Line 48: delete either 'recording' or 'documenting' (one verb too many).

*Line 72 – Removed 'recording'*

Line 131: 'European Centre for Medium Weather Range Weather Forecast'. Delete the first 'Weather' as it should say 'European Centre for Medium-Range Weather Forecast'.

*Line 95 – Removed the first mention of 'Weather'*

Line 198: different => difference

*Line 233 – Adjusted to difference as requested*

Line 199: complied => compiled

*Line 236 – Changed to compiled*

Line 282: You refer to a paper by Taylor et al. (2022), yet this paper is not listed in the reference section.

*Line 475 – Taylor et al (2022) has been added to the references.*

Line 336: Please add the name of the journal where this article was published.

*Line 387 – Journal name (Terrestrial Photogrammetry) has been added to the Brecher and Thompson (1993) reference.*

Lines 353-354: delete 'an international journal'.

*Line 406-407– Deleted ' an international journal' in Hall and Riggs (2007) reference.*

Lines 335-339: You repeat the same reference twice. Delete one of them. Also, in the text simply cite Hanshaw & Bookhagen (2014). There is no need for the labels 'a' and 'b' – it's one and the same paper.

*Line 408 – We removed the extra reference. All in text citations should now read Hanshaw & Bookhagen (2014) on Lines 20, 108, and 157.*

Line 382: delete 'Article 3'

*Line 438 – Deleted 'article 3' as requested in Mark et al (2002) reference.*

Line 388: This should be Pepin 'et al.' (the paper has many co-authors). Also note that there is an updated newer version of this paper (Pepin et al., 2022).

*Line 448 – Adjusted Pepin et al., (2015) reference. Also fixed in text citation (Line 21). Added Pepin et al., (2022) reference as well in both text (Line 21) and in reference list (Line 445).*

Line 404: delete 'Article 1-5'.

*Line 477 – Deleted 'Article 1-5' as requested in Thompson et al (2000) reference.*

Line 433: delete 'Article 9'.

*Line 506 - Deleted 'Article 9' as requested in Vuille et al (2015) reference.*

References:

Hurley, J.V.,  et al. 2015. Cold air incursions, d$^{18}$O variability and monsoon dynamics associated with snow days at Quelccaya Ice Cap, Peru. *J. Geophys. Res*., 120, 7467-7487, doi:10.1002/2015JD023323.

Hurley, J.V.,  et al. 2019. On the interpretation of the ENSO signal embedded in the stable isotopic composition of Quelccaya Ice Cap, Peru. *J. Geophys. Res.* 124, 131-145, doi: 10.1029/2018JD029064.

Kochtitzky, W.H., et al. 2018. Improved estimates of glacier change rates at Nevado Coropuna Ice Cap, Peru. *J. Glaciol*., 64(244), 175-184, doi: 10.1017/jog.2018.2.

Pepin, N.C., et al. 2022. Climate changes and their elevational patterns in the mountains of the world. *Rev. Geophys*. 60, e2020RG000730, doi:10.1029/2020RG000730.

Rabatel, A., et al., 2012: Can the snowline be used as an indicator of the equilibrium line and mass balance for glaciers  in the outer tropics? *J. Glaciol.,* 58(212), 1027-1036. doi:10.3189/2012JoG12J027

Rabatel, A.,  et al. 2013. Current state of glaciers in the tropical Andes. A multi-century perspective on glacier evolution and climate change. *Cryosphere*, 7, 81-102, doi:10.5194/tc-7-81-2013.

Sulca, J., et al.  2018. Impacts of different ENSO flavors and tropical Pacific convection variability (ITCZ, SPCZ) on austral summer rainfall in South America, with a focus on Peru. *Int. J. Climatol.,* 38, 420-435, doi:10.1002/joc.5185.

Yarleque, C., et al. 2018.. Projections of the future disappearance of the Quelccaya Ice Cap in the Central Andes. *Sci. Rep.* 8, 15564, doi:10.1038/s41598-018-33698-z.

---

## Author Comment (AC2)

**Response to Reviewers**

*We thank both reviewers for their constructive feedback. We have addressed all major and minor comments below, our responses are in blue text. Specifically, we adjusted the topographic correction within the analysis code and adjusted our target date for imagery based on comments from the reviewers. All results have been updated as well as the corresponding figures both in text and in the supplement. A note that Figure 5 is now Figure 6 in our revisions. We were informed that the colormap on Table 1 would not be possible to do with color shading so we have created a bar chart figure instead, which is now Figure 5.*

*Reviewer #2*

This manuscript evaluates the change in the Quelccaya Ice Cap from 1985 to 2023, using satellite imagery. The year-to-year changes are described, and an evaluation is conducted as to the change in strong El Niño years. The analysis is interesting, particularly the discussion of the snow covered area rebound time after El Niño events.

Some clarification is needed before publication, particularly on the definition of El Niño years and some of the statistical analysis. Detailed comments are provided below.

**Comments**

Full manuscript: Could the authors clarify which El Niño years are chosen for which parts of the analysis in this manuscript, and why? The authors focus on the 1997-1998, 2015-2016 and 2023 events for much of the manuscript, but use more years for figure 5, for example. Some graphs (e.g. S4) are unclear in which ENSO years are used.

*Line 215 – We note that the strongest El Niño events are the ones considered for our analysis. For figure 6 (previously 5), we chose to display all Ninos and Ninas with an ONI index that surpasses 1.0 or -1.0. Clarifying language has been added to the discussion section to reflect that decision (Line 311 and Figure 6 caption).*

*Line 312-315- Figure S4 – We added details to the discussion surrounding Figure S4. It was created as an assessment across the entire observed temporal scale to evaluate how the measured variables change with response to neutral vs ninos vs ninas. These events were used based on the previously mentioned ONI index threshold that we have defined, and has been clarified (Line 311).*

Line 86 and Table S1: The authors have picked the closest cloud-free images to September 1$^{st}$ in order to calculate the area of the ice cap accurately, however these images vary from June to October. Can the authors comment on the effect of this difference on their results?

*Line 114-122: We have adjusted our window for imagery selection to better target the end of the dry season. The only July imagery is from the first two years as no other options were available, and we clarified our reasoning for how and why we chose and acquired imagery. The section now reads 'To automate the SCA detection and ELA calculation, the following data inputs were required: an annual satellite image, a DEM, and the 1985 outline of the QIC. Using the Google Earth Engine platform (GEE) we selected annual Landsat images as close as possible to September 15$^{th}$ with clear visibility of the QIC from 1985 to 2023 (Table S1). Mid to end September marks the end of the dry season in the CV, which enabled analysis of the ice cap without extraneous snowfall around the perimeter. Imagery from each year was on average ±23 days within the target date and was manually inspected to ensure no*

*recent snowfall events occurred. If September imagery was not available, October and November images were collected, and if imagery was still not available August and July were collected with the intention to collect the closest to end of dry season conditions at the QIC. No images were used if a recent snowfall event was evident. Sentinel-2 imagery was used in 2021 and 2023, due to a lack of cloudless images from Landsat 8/9.'*

Line 96: 1999 was a strong La Niña year, not an El Niño year (as the authors note in line 261). 2017 was a neutral year. Can the authors clarify the dates of the 16 and 18 images collected, and what the ONI index was for the rolling three month period covering those dates?

*Line 128-130 – Dates clarified for the 1998 and 2016 El Nino event sampling.*

*Line 308 – The ONI index during 1999 is now noted in this discussion section regarding the rebound of the SCA.*

Line 126: Could the authors expand on the difference between the manual calculation of ELA and the automated calculation? Is the 3 % in this instance 3 % of the elevation above sea level? If so, this would be greater than the change in ELA between 1985 and 2022 that the authors calculate on line 143, and may require further consideration.

*Line 157-160 – Our discussion of the error was centered around the manual tracing of the SCA&TA and not the ELA. Apologies for the typo. We have corrected and clarified the language as to the error calculation and included an additional reference. Additionally, we have updated Figures 2, 3, and 6 (previously 5) to include error bars/semi-transparent margins of error on the line graphs.*

*The new section includes a new reference as well to further assist in explanation and now reads 'Calculated results for the SCA and TA via our automated methods are in good agreement with manual digitizations (within ±3%). Other studies have shown manual and automated detection of snowline produces similar result to manual digitization and low level of error (Hanshaw & Bookhagen, 2014) with automated detection being preferable to manual as repetition is simpler and any error is likely to be more consistent (Paul et al., 2013)*

Line 142-146: The authors have calculated the loss of TA and SCA based on the first and last years of data only. Especially for SCA, the year-to-year variation is relatively high. I would suggest the authors calculate the loss through the average loss (which would then match with the loss calculated through linear regression), or take the average of the first few and last few years. This comment also holds for figure 2.

*Line 173 – We have now included average loss per year in our results.*

*Figure 2 – The intention of this figure was to display the shift of the snow cover area (SCA) to higher elevations. We would prefer to leave it as is, since the intent was to display the drastic change over the short time period. Error bars are now included, see above. We added some more information into the results section discussing average shift from the first five to the last five years, as suggested, immediately following the full scale results. Lines 173-175 reads 'Over the observation period (1985 and 2022), the QIC lost ~37% of its TA and ~58% of its SCA (1985: TA=~58.7 km$^2$, SCA=~46.3 km$^2$; 2022: TA=~36.7 km$^2$, SCA=~19.7 km$^2$ (Table S2).Average loss between first five observed years and the last five recorded an SCA decline of ~38% and the TA by ~29%, respectively.'.*

Line 154: Could the authors clarify why they have removed El Niño years to calculate the trend in AAR, but not comparative La Niña years? The analysis shown in figure S4, which clearly shows the

difference between El Niño, La Niña and neutral years, may be more appropriate to determining the effects of El Niño events on the AAR.

*Line 184-187 – We have amended the average AAR to only include neutral years instead of neutral and La Nina years, and added La Nina years to read 'The QIC's average AAR (minus El Niño and La Niña years) is 0.74 over the study period. Conversely, during the strongest El Niño years (1998, 2016, and 2023) the QIC's AAR was 0.32, 0.40, and 0.52, respectively and during the strongest La Niña years (1999 and 2011) the QIC's AAR was 0.83, and 0.82.'*

Figure 5: The x-axis seems to be misaligned in this plot. For example, the lowest SCA occurred in 1998, along with the El Niño year. Both of these are plotted covering the tick mark for 1999.

*Figure 6 (previously 5) – Line 328– The Nino and Nina bars were offset unintentionally. We had adjusted them accordingly to cover the intended year. See figure change above in reviewer #1 comments.*

Line 163: I agree with the other reviewer comment that it is necessary to apply a simple bias-correction to the ERA5 temperature, to account for the difference in elevation between the QIC and the ERA5 elevation, in order to determine the percentage of days above the freezing level height. I would also suggest that the authors' findings of an increase of 1.24 degrees in the dry season over 37 years is somewhat greater than the 0.1 degree/decade in the quoted references.

*Line 188-190– As mentioned above under reviewer #1s comment, we have assessed and readjusted to the 550 mb ERA5 temperature as it most properly reflects the observed temperature at the QIC (e.g., Bradley et al., 2009). In text changes have been adjusted accordingly to read 'Daily and monthly variations recorded by the QIC summit and bottom margin weather stations from Bradley et al., (2009) are well correlated with the ERA5 550mb temperature dataset, which was analyzed to determine changes in climatic variables through the observation period.'*

Line 187 and figures 2 and 3: These figures might be easier to interpret if they were plotted the total SCA in each elevation bin, rather than the percentage. For example, in figure 3 (lower), thepercentage of SCA in 2015 and 2017 is similar at low elevations, suggesting the SCA may be similar in both years, but they vary considerably at higher elevations. Total SCA would then match better with the text stating the change in SCA is 70% of its 2015 value.

*We chose to use the percentage of the SCA in Figure 2 so the comparison between the beginning and end of the temporal scale is normalized and the shift to higher elevations is evident. We have altered Figure 3 as suggested to display the total SCA values since, in this case, we show consecutive groups of three years.*

Line 253: Lagos et al., 2008 record higher than average precipitation during El Niño events along the northern coast, but a mixed response in the Amazon and the Andes (including around the QIC).

*Line 298-299– Reviewer #1 noted a similar concern with this sentence, we have adjusted accordingly and added their proposed references. It now reads 'During El Niño events the Peruvian Andes are often drier than average (Sulca et al., 2018), with on-site measurements at Quelccaya recording warmer and drier conditions (J. Hurley et al., 2019).'*

Line 263: Could the authors explain how they have calculated the drop in correlation from 1998 to 2016? Are these correlations based on the yearly measurements and yearly ONI? If so, how is the correlation calculated for a single year? Similarly, how are the regressions calculated in line 266 (and

what are the multiple variables used?.

*Line 220 -222 – We have clarified this process by changing the text to read 'To better determine QIC changes during the El Niño events, high frequency sampling was conducted around the complete El Niño events, consisting of 16 and 18 images collected between 1997–1999 and 2015–2017, respectively.' The results are reported onLine 224-225– The correlation is calculated using the high frequency sampling previously discussed to analyze changes during the El Ninos. We have added clarifying language to discuss that we evaluated those images during when the three month avg ONI recorded Nino conditions. The adjusted sentence reads In addition, correlation between the monthly ELA and monthly ONI index during the two El Niño events (1997-1999 and 2015-2017) are 0.68 and 0.26, respectively The referred to Line 266 has been deleted in effort to clarify the prior comment and we move on to discuss El Nino as a binary predictor in the regression.*

Line 265 and 297: the authors state that anthropogenic warming is overwhelming natural climatic signals, but in this manuscript, the most severe El Niño effects seem larger than the overall reduction from anthropogenic warming (e.g. figure 5, 1998/1999)

*Line 315-321 – We clarify this statement that while the QIC responds to ENSO fluctuations in the most recent La Nina (2021/2022) the SCA only declines and in all the others it experienced some level of rebound. We added additional commentary about the effects of anthropogenic warming and while this lack of rebound is a single isolated event, we expect this behavior to continue into the future with the next La Nina. The new section reads 'While the SCA is notably briefly impacted by these El Niño events, decline from anthropogenic warming has resulted in the long-term decline of the SCA and TA of the QIC (Bradley et al., 2009; Rounce et al., 2023; Thompson et al., 2021; Vuille et al., 2018; Yarleque et al., 2018). Further, during all previously noted La Niña events (Fig. 6), the SCA experienced some level of temporary expansion, but throughout the 2021-2022 La Niña, the SCA did not rebound, but only declined further. While this is only one incidence, we expect this behavior to continue through the onset of the predicted upcoming 2024/2025 La Niña. The decrease in the percentage of days at or below freezing during the wet season will only exacerbate the decline in SCA.'*

**Minor suggestions**

Line 9: I would suggest rephrasing the first sentence to "Tropical glaciers in the central Andes are vital water resources….". As it stands, water resources are mentioned twice.

*Line 9 - We have adjusted this sentence as requested and removed the double mention of resources.*

Line 30: Please defined what the modern rate is of (retreat, volume loss etc). and Line 30: 'further evidence of the QIC …' what's this further evidence of, is it retreat?

*Line 33-34 – We have adjusted this sentence to clarify ice margin retreat and the QIC's evidence of past fluctuations. It reads 'Further evidence of the QIC's past fluctuations has recently been placed within a longer-term context using radiocarbon-dated plant remains from the ice margin suggesting that the ice cap's present-day margin extent has not occurred in the last 7,000 years (Lamantia et al., 2023).'*

Line 32: 'since 7000 years ago' -> 'in the last 7000 years'

*Line 34: Adjusted the phrasing as requested. See above comment.*

Line 42: 'southern wet outer tropics' 'wet' is perhaps a typo here?

*This sentence was removed and replaced by discussion of the SAMS (see reviewer #1 comment)*

Line 48: typo 'recording documenting'.

*Line 72 - Removed 'recording'*

Line 114: Please define 'Ab'

*Line 144 – Ab has been defined in the text. Ac = accumulation area (snow). Ab = ablation area (ice). Ac+Ab = Total Area*

Line 152: as the AAR is defined based on the SCA and TA, I think the correlation between these variables seems inevitable.

*Line 195. This is a great point, we have removed the reference to AAR correlation and the corresponding supplemental table.*

---

## Referee Report (RR1)

This manuscript has been greatly improved, however there is still some confusion and imprecision about the definition of El Niño events which should be rectified before publication.

**Comments**

Line 200: Greater description of the ENSO indices used (with webpage references, if this is where the three indices are taken from) would help clarify the subsequent analysis. Line 200-201 is somewhat confusing, is there a lagged correlation between ONI indices and variables, and if so, is it months or one year? Or are the authors determining years to be considered El Niño or La Niña based on some threshold of number of months (and if so, what month is taken as the year start)?

Line 150, 264, 268 and section 3.2: As noted, 1999 and 2017 contain no El Niño periods so cannot be defined as El Niño years, and defining 1997-1999 and 2015-2017 as El Niño events is therefore incorrect, and negates many of this paper's concluions. 1999 was a strong La Niña year, as noted elsewhere in this manuscript. Please check ENSO dates are precise throughout the manuscript, using e.g. https://origin.cpc.ncep.noaa.gov/products/analysis_monitoring/ensostuff/ONI_v5.php .

In relation to this, section 3.2 refers in the first paragraph to the strong El Niño events of 1998, 2016 and 2023. The analysis of the first two events is consistent, as they both consist of strong El Niño indices over the preceding wet season, until around March/May of 1998/2016. 2023, however shows the opposite pattern, with strong La Niña indices over the preceding wet season. Given that the strongest El Niño months for 2023 occurred after the date of the satellite imagery from which this analysis has been used, I do not think it is reasonable to make conclusions about the effect of the 2023 El Niño at this point, unless all analysis has been conducted using only a few months of ENSO index leading up to the date of satellite image.

Figures throughout: Please define in the figure captions the meaning of all error bars and shading, and which data were used to construct these.

**Minor comments**

226: Please use SI units hPa rather than mb

265-267: The statement that there is a steady decline over 3 consecutive years from both 1997-1999 and 2015-2017 contradicts lines 259-260 and figure 3, which show a rebound or partial rebound in the final year.

Line 278 and Table S4: which El Niño years are included here?

Table S5: MEI mislabelled as MVI.

---

## Author Response (AR2)

**Response to Reviewers**

*We thank the reviewers for their comments. All responses are below in blue text. Line numbers correspond to the changes accepted document.*

Reviewer #1

This manuscript has been greatly improved, however there is still some confusion and imprecision about the definition of El Niño events which should be rectified before publication.

**Comments**

Line 200: Greater description of the ENSO indices used (with webpage references, if this is where the three indices are taken from) would help clarify the subsequent analysis. Line 200-201 is somewhat confusing, is there a lagged correlation between ONI indices and variables, and if so, is it months or one year? Or are the authors determining years to be considered El Niño or La Niña based on some threshold of number of months (and if so, what month is taken as the year start)?

*Lines 173-174 - Clarified the collection of the indices from the NOAA database, and provided this information and a link to the data.*

*Line 176-178 – Yes there is a lagged correlation where the result is correlated back to the indices for each month over the duration of the previous year. We edited the text to clarify this process for the reader a bit more. It now reads 'As the average month of observation for each year was September, the result variables for each year were correlated with the preceding months' indices, one year before the annual September observation date (e.g. 1998 SCA is correlated with the indices starting in August 1998 and going backward to September 1997.' Results are shown in Figure 5.*

Line 150, 264, 268 and section 3.2: As noted, 1999 and 2017 contain no El Niño periods so cannot be defined as El Niño years, and defining 1997-1999 and 2015-2017 as El Niño events is therefore incorrect, and negates many of this paper's conclusions. 1999 was a strong La Niña year, as noted elsewhere in this manuscript. Please check ENSO dates are precise throughout the manuscript, using e.g. https://origin.cpc.ncep.noaa.gov/products/analysis_monitoring/ensostuff/ONI_v5.php .

*Section 3.2 -We did not intend to identify 1997-1999 as El Niño events. Line 239-241 states 'To better assess the QIC's response to El Niño events, we utilize our high-frequency (monthly) observations collected around the 1998 and 2016 El Niño events (i.e., between 1997–1999 and 2015–2017).' The intent was to analyze and discuss how the QIC responded to the El Niño events, and thus we needed data from before and after, aka the 1997 and 1999 measurements. From the previous comment, we also defined the indices as Nino or Nina qualifying (Line 174-175).*

In relation to this, section 3.2 refers in the first paragraph to the strong El Niño events of 1998, 2016 and 2023. The analysis of the first two events is consistent, as they both consist of strong El Niño indices over the preceding wet season, until around March/May of 1998/2016. 2023, however shows the opposite pattern, with strong La Niña indices over the preceding wet season. Given that the strongest El Niño months for 2023 occurred after the date of the satellite imagery from which this analysis has been used, I do not think it is reasonable to make conclusions about the effect of the 2023 El Niño at this point, unless all analysis has been conducted using only a few months of ENSO index leading up to the date of satellite image.

*Line 233-238 – We added clarifying language to the results to indicate that while the El Niño in 2023 was not at its maximum, it was in fact in existence a few months before the 2023 measurement occurred. We could not measure it later in 2023 due to cloud cover and the consistency of our previous measurements.*

*Discussion about the differences surrounding these events is in the Discussion, Section 4.1. Specific discussion about the 2021-2022 La Niña begins at Line 324.*

Figures throughout: Please define in the figure captions the meaning of all error bars and shading, and which data were used to construct these.

*Figure 2, 3, & 6 are applicable to this comment. We have added the following to each caption 'Error bars represent ±3% uncertainty calculated from comparisons to manual digitization.'*

**Minor comments**

226: Please use SI units hPa rather than mb

*Line 198 – Changed to hPa as requested*

265-267: The statement that there is a steady decline over 3 consecutive years from both 1997-1999 and 2015-2017 contradicts lines 259-260 and figure 3, which show a rebound or partial rebound in the final year.

*Line 240-242: We adjusted this statement as we are discussing the decline into the El Niño year. It now reads 'We found that in both the 1997–1999 and 2015–2017 periods, the lowest SCA occurred during the El Niño years during the mid-September observation and that the decline of the QIC's SCA began from the previous year's September measurement.'*

Line 278 and Table S4: which El Niño years are included here?

*Line 278 – This is the middle of Figure 3, the years are available in the figure legend. To reference Table S4, we have updated the caption to reflect the usage of an ONI index ±1.0 to classify the years as El Niño, La Niña, or Neutral. This is also discussed in Lines 321 where Table S4 is mentioned and we have clarified the mention of Table S4 in the text in Lines 257.*

Table S5: MEI mislabelled as MVI.

*Table S5 – Label changed to MEI*

Reviewer #2

The authors improved the paper and addressed the main issues. Hence I have no further major comments. However, there are still many small errors interspersed throughout the paper, so I would urge the authors to thoroughly proof-read the paper one more time before publication. I listed a few mistakes below that caught my eye.

Line 66-67: Why include initials in this citation? Simply write: '(Hurley et al., 2015).'

*Line 69 – citation adjusted to Hurley et al., 2015.*

Line 66-67: Same comment – remove initials from citation: '(Thompson, 2017).'

*Line 72 – Citation adjusted to Thompson, 2017.*

Line 72: 'Additionally, ice cores from multiple locations in Peru document this accelerating enrichment'. What enrichment? This sentence is not clear.

*Line 74-77 – Enrichment of δ¹⁸O. Have clarified the sentence to read 'This warming trend is also reflected in ice core stable isotope (δ¹⁸O) records from multiple locations in Peru (Thompson, 2017; Thompson et al., 2013, 2017). High-resolution ice core records indicate that the QIC is an excellent recorder of El Niño, characterized by elevated sea surface temperatures (SSTs) in the Eastern Pacific Ocean, with strong events recording isotopically enriched δ¹⁸O (Thompson et al., 2011, 2017).'*

Line 95-96: This is still wrong. You write: 'European Centre for Medium Weather Range Weather Forecast (ECMWF), yet it should say: 'European Centre for Medium- Range Weather Forecast (ECMWF)'.

*Line 98-99 – Adjusted acronym definition to 'European Centre for Medium-Range Weather Forecast (ECMWF).'*

Line 148: This is the first time you use the abbreviation 'OTSU. Please spell out what this means.

*Line 140 – OTSU is not an abbreviation, but the surname of the scientist who developed the method. We have altered the sentence to read, 'To delineate the snow cover area (SCA), the NIR band was assessed with an image segmentation algorithm, the Otsu method (Gaddam et al., 2022; Otsu, 1975).' This is to clarify this statement and added the original reference from Otsu's publication.*

*We also mention it on Lines 116, 154, &155 and have adjusted the wording there as well.*

Line 143: This is section 2.4, not 2.3.

*Line 147 – Changed heading to 2.4*

Line 148: This is the first time you use the abbreviation 'SWIR'. Please spell out what this means.

*Line 152 – Defined SWIR in text as short-wave infrared*

Line 161: This is section 2.5, not 2.4.

*Line 165 – Changed heading to 2.5*

Line 196: these r-squared values are very low – are you sure the significance levels you indicate for such low vales are correct? How can a $r^2$ value of 0.03 be significant at p=0.01?

*Line 205– We re-checked and adjusted the p values. The r2 values are low and the p values are not significant as stated in the text and reflected by the reported results.*

Line 226: ERA 5 data show (plural)

*Line 259 – Changed shows to 'show'*

Line 245: p-values are usually indicated with a small 'p'. You do so in the first part of the paper, but here you suddenly start to capitalize 'P'. Be consistent, as otherwise one might come to believe that

you are referring to two different metrics or variables. The same comment applies to the seemingly indiscriminate use of 'r' and 'R' throughout the paper.

*Line 251 – p value has been changed to lowercase. All of notations of p values were checked to ensure consistency.*

Figure caption 3: you write 'Percentage of snow cover..', yet the Figure shows absolute snow cover in km^2.

*Figure 3 caption has been changed to 'Distribution of Snow Cover (km2) instead of percentage.*

Figure caption 4: You write: 'Decrease in the QIC' SCA (red) and TA (blue) at the end of the dry season from 1985 (left) to 2023 (right)'. Yet you only show the decrease in the SCA. The TA (blue line) is the same in both images.

*Figure 4 has been adjusted to show the decrease in total area and snow cover area in both years.*

Line 299: Remove initials. Simply write: '(Hurley et al., 2019).'

*Line 303 – reference changed to Hurley et al., 2019*

Line 315-316: this sentence is unclear: 'decline from anthropogenic warming has resulted in the long-term decline of the SCA'. What do you mean with 'decline from anthropogenic warming'?

*Line 317 – This is a confusing sentence, we have altered it to read 'This indicates that while the QIC's SCA is notably briefly reduced, and its decline exacerbated over the long term, by El Niño events, anthropogenic warming is the primary driver of the multi-decadal decline of the QIC's SCA and TA (Bradley et al., 2009; Rounce et al., 2023; Thompson et al., 2021; Vuille et al., 2018; Yarleque et al., 2018).'*

Figure S5: I don't find such simple linear interpolations to be very useful or realistic assumptions. As ice caps shrink and become thinner, elevation-dependent feedbacks and edge effects will become increasingly more important, resulting in accelerated shrinking over time, especially given the large flat topography making up most of the remaining ice-covered area.

*Lines 350-355 – We used Figure S5 to discuss this point, that if we assume constant loss that's what we'd expect to see happen, but we note there is potential for many other factors to influence this loss and it will likely not be a linear decline. We want to include figure S5 for the sake of future discussion as the QIC's total decline has been so close to linear for decades, how will it change when it becomes non-linear? There is a good potential for more studies into what caused that shift in its decline. We do note this in the text as well, that linear decline is unlikely.*

Line 347: 'El Nino': add the 'tilde' symbol above the 'n'.

*Line 344 – Tilde added above 'n'*

Lines 387-388: This reference is still wrong. It lacks volume and page numbers or doi. Also the journal name is wrong. This article appeared in the journal 'Photogrammetric Engineering & Remote Sensing', not 'Terrestrial Photogrammetry'. Finally, the title is incomplete: The correct title is: 'Measurement of the Retreat of Qori Kalis Glacier in the Tropical Andes of Peru by Terrestrial Photogrammetry'.

*Lines 398-399 – We have adjusted the reference to include volume and page numbers as well as correct the journal. There is no doi as the paper is only hosted on the Byrd center webpage as a scanned copy of the document or a hard copy.*

Lines 446-448: This reference is missing half the authors. Please include the complete author list or abbreviate the author list in a way that makes it clear that not all authors are listed.

*Lines 458-459 – This reference (Pepin et al, 2022) has been checked to list all authors and missing ones added to the reference listing.*

Supplement: Please adjust the manuscript title. You are still using the old erroneous title.

*Title adjusted to match manuscript*

Supplemental Table 2: It makes no sense to indicate median elevation and ELA with an accuracy of millimeters. I think meters would be a better reflection of the uncertainty of these estimates.

*Supp Table 2 – We have adjusted the ELA and median elevation to an accuracy of meters as suggested.*

---

## Author Response (AR3)

*We thank the editor for their feedback. All our responses are in blue text below.*

I am pleased to accept your manuscript for publication in The Cryosphere subject to the following minor revisions

1. Please replace references to ENSO as a 'short-term' climate mode with 'interannual' climate mode.

*We have replaced the ENSO references with interannual instead of short-term on Lines 10, 83, 100, 131, 214, 286, 328, and 344.*

2. For ease of reading, please reduce the number of abbreviations where possible. For example Cordillera Vilcanota (CV) and region of interest (ROI) are only used twice in the whole manuscript and do not need to be defined as an abbreviation.

*Yes, there are many abbreviations, and unfortunately many are unavoidable. As suggested, we have removed the ROI and CV acronyms on Lines 30, 120, 135, and 137. We also removed ECMWF from lines 98 and 168, SRTM on 126, and GEE from lines 118, 356, and 37#.*

3. Please provide additional information about how the agreement with manual digitisation and the ±3% error estimate was assessed (based on how many/which image comparisons, etc).

*Line 161-164. We have clarified the range of error came from comparing TA and SCA computed with our automated methods with values from manually digitized values from 10 yearly images, and we list the specific years of the imagery used.*

4. If I understood correctly, you identified El Niño and La Niña events based on single monthly values of the ONI index, however traditionally they are defined based on the five consecutive 3-month running mean of SST anomalies in the Niño 3.4 region exceeding the threshold of ±0.5°C (see https://www.ncei.noaa.gov/access/monitoring/enso/sst). Please justify, assess, and discuss the impact of your approach.

*We chose to evaluate the ONI on a month-to-month basis as we were correlating our results back each month to better evaluate the response the QIC exhibited from strong El Niños and La Niñas. By setting the threshold at 1 instead of 0.5 we focus on the stronger El Niño events that have noticeably impacted the ice cap over time. For example, there are smaller El Niño events (2002, 2004, 2006, 2018/2019) that do not feature an ONI index over 1.0. While still considered El Niño events they are not as aggressive and did not produce the stark changes that we observed in the 1998 and 2016 El Niños. Clarifying language has been added to Lines 176 - 181.*

5. For clarity and ease of reading, please provide a consistent end date of your study throughout the manuscript. The 2023 El Niño event is considered and frequently discussed, however in Section 3 you refer to "the observation period (1985 to 2022)."

*Line 186, we have changed the beginning of section 3 to read 'Between 1985-2022' instead of 'the observation period.' We added 'this observation' to the following sentence to clarify this as well (Line 187). We explain in section 3.2 beginning on Line 221 that the 2023 measurement is included as an insight into El Niño since the measurement occurred during an ongoing event. We add clarifying*

*language on Line 193-195 as well regarding the linear regression results and removal of El Niño years.*

6. Line 197-199: where can the reader see the agreement between ERA5-Land and station data?

*Line 202 - We have provided the correlation coefficient in the text now to clarify the dataset agreements. This statement was originally added because a reviewer asked if we could explain why we chose to use 550 and not surface temperature or another pressure level, etc. The station data is available in the Bradley et al, 2009 reference and ERA5 Data is publicly available.*

7. Line 223-225: "We found that in both the 1997–1999 and 2015–2017 periods…" I still find this sentence hard to follow, please simplify and clarify if possible

*Line 230 - 234 - We have altered this sentence to read 'We found that during both the 1997–1999 and 2015–2017 periods, the lowest SCA occurred during the mid-September observations of El Niño year (1998 and 2016), after a decline in SCA that began from the previous year's September measurement (1997 and 2015). The ENSO indices are most strongly correlated with the QIC's ELA, SCA, and median elevation as they best represent the changing ice distribution and mass.'*

8. Line 226-228: please move to the text justifying only discussing the ONI index to the methods section 2.5

*We have moved this text to the methods section, it is now on Line 181-182.*